# Shear Performance of Reinforced Concrete Beams Affected by Satisfactory Composite-Recycled Aggregates

**DOI:** 10.3390/ma13071711

**Published:** 2020-04-06

**Authors:** Changyong Li, Na Liang, Minglei Zhao, Kunqi Yao, Jie Li, Xiaoke Li

**Affiliations:** 1International Joint Research Lab for Eco-building Materials and Engineering of Henan, North China University of Water Resources and Electric Power, Zhengzhou 450045, China; liangna2215@163.com (N.L.); bgyykq2018@163.com (K.Y.); 2School of Civil Engineering and Communications, North China University of Water Resources and Electric Power, Zhengzhou 450045, China; lixk@ncwu.edu.cn; 3School of Engineering, RMIT University, Melbourne, VIC 3003, Australia; Jie.li@rmit.edu.au

**Keywords:** composite-recycled aggregate concrete (CRAC), reinforced concrete beam, stirrups, shear strength, shear-crack width, shear capacity, utilization level of CRAC

## Abstract

This paper is the outcome of experiments on the shear performance of reinforced concrete beams with approved composite-recycled aggregates. The strength grade of composite-recycled aggregate concrete (CRAC) was between 30 MPa and 60 MPa. The shear span-to-depth ratio varied from 1 to 3. The adaptability of HRB400 rebar, with critical yield strength of 400 MPa, used as stirrups was also verified. As the composite technology overcame the shortcomings of recycled coarse aggregate, CRAC had similar mechanical properties with those of conventional concrete. Details on the shear behaviors of test beams under a four-point loading test are presented. The results indicated that the changes of CRAC strain, stirrup strain, and shear-crack width depended on the failure patterns, which are controlled by the shear-span to depth ratio. The stirrups yield at the failure of reinforced CRAC beams. The shear cracking resistance and the shear capacity of reinforced CRAC beams can be predicted by the statistical equations. Based on the design codes GB50010, ACI318-19, Model Code 2010 and DIN-1045-1-2008 for conventional reinforced concrete beams, the shear strengths provided by CRAC and stirrups are statistical analyzed. The rationality of the design equations is examined by the utilization level of shear strength provided by CRAC. The maximum shear-crack widths are extracted from the test data of reinforced CRAC beams at normal service state. Comparatively, by specifying the lower limit of shear strength provided by the CRAC with various shear-span to depth ratios, China code GB50010 gives a rational method for utilizing CRAC. Under the premise that the design of shear capacity would give considerations to meet the normal serviceability, the factored strength of HRB400 rebar should be 360 MPa for the calculation of shear strength provided by stirrups. The design methods in codes of GB50010, ACI318-19 and Model Code 2010 are conservative for the shear capacity of reinforced CRAC beams.

## 1. Introduction

To substitute the natural aggregates of concrete with the recycled aggregates made of mining waste and demolished concrete structures, different approaches have been investigated in recent decades [1,2,3]. Efforts have been put into improving the recycling process and technology of recycled aggregates to eliminate the defects of recycled coarse aggregate, including irregular morphology of particles, and rough surface bonded with old mortar leading to greater water absorption [4,5,6,7,8]. The mix proportion design of concrete has been adjusting by using mineral admixtures [9,10], and different approaches of recycled aggregates replacing natural aggregates. These approaches include coarse natural aggregate being partially replaced by recycled coarse aggregate, natural sand partially or all replaced by recycled fine aggregate, and full use of recycled fine and coarse aggregates [2,6,11,12]. The absolute volume method has been generally accepted in the design of mix proportion to deal with the difference of density among raw materials. The mixing procedure attains perfection by adding the step of pre-soaking the recycled aggregates with additional water, or pre-wrapping recycled coarse aggregates with a certain amount of cement paste [10,11,12,13,14]. Different properties of concrete with recycled aggregates are given out due to the different approaches applied in the investigations. In order to maintain a safe and economical application, technical specification on recycled aggregate and recycled aggregate concrete have been introduced [15,16,17,18].

Following the worldwide research steps, the authors’ research team draw inspiration by absorbing the successful experiences in the investigations of recycled aggregate concrete. The purpose is to fully use the recycled fine and coarse aggregates without byproduct emission [11,12,19,20,21,22]. A three-stage-recycling-technology is formed to produce recycled aggregate with rational morphology. The demolished concrete blocks and the secondary crushed particles are crushed by the jaw crusher. Particles are then shaped and graded by the vertical impact crusher and a screening system. To remove the old mortar bonded on the surface of recycled coarse aggregate, recycled coarse aggregate is crushed into particles with smaller sizes than that of the coarse aggregate of demolished concrete. For instance, if the old concrete is demolished from structures with mother particles of 25mm, the recycled coarse aggregate will be produced with particle size less than 20mm. In this case, new concrete can be made of full-recycled coarse aggregate in continuous grading with a maximum particle size of 20mm, accompanied with recycled fine aggregates [22,23]. However, in many cases, the demolished concrete comes from multiple sources with different structures. Particle size of the mother aggregate is not easy to be certified. Therefore, the recycled coarse aggregate is usually produced to be particle with a maximum size of 16mm. In this case, to meet the requirement of a continuous grading coarse aggregate [24], a certain amount of natural coarse aggregate with particle size no less than 16mm is added. This forms a composite-recycled aggregate. Combined with the perfecting of the mixing process, a new concrete with composite-recycled aggregate (abbreviated as CRAC) is produced [11]. As the composite technology overcomes the shortcomings of large particle recycled coarse aggregate with rough surface attached old mortars, the mechanical properties of CRAC are similar to those of conventional concrete [25,26]. Based on the experimental investigations, the reinforced CRAC columns under eccentric compression and beams in bending exhibit the same loading behaviors compared with those of conventional concrete [27,28]. As part of the experimental studies, the shear performance of reinforced CRAC beams with stirrups was planned to be conducted. The results are reported in this paper.

In this aspect, previous researches on shear behavior of reinforced recycled aggregate concrete beams can be used as references. Comparisons were made with conventional reinforced concrete beams. For reinforced recycled aggregate concrete beams with no stirrup and shear-span to depth ratio of 2.6, 3.0 and 3.0, negligible effect was found on shear cracking pattern, critical shear cracks, longitudinal reinforcement strain, and failure mode, except for the 14%, 9% and 12% reduction of shear capacity, due to the use of 100% recycled coarse aggregate with maximum particle sizes of 20, 19 and 25mm, respectively [29,30,31,32]. Meanwhile, less effect was found on the shear performance of reinforced recycled aggregate concrete beams with stirrups and shear-span to depth ratio of 3. This was mainly due to the lower tensile strength of recycled aggregate concrete than that of conventional concrete with an equal compressive strength [31]. With the same compressive and tensile strengths of recycled aggregate concrete compared to conventional concrete, the deflections and ultimate loads of beams produced with stirrups ratio from 0.21% to 0.5% and shear-span to depth ratio of 3.3, are little affected by the concrete types, except the shear cracking is premature in recycled aggregate concrete beams [33]. Adding 8% silica fume in the weight of the cement to the recycled aggregate concrete could improve the shear cracking of reinforced recycled aggregate concrete beams, while the ultimate loads tend to be the same [34]. When the concrete with coarse recycled aggregate replacing 50% and 100% natural aggregate has similar compressive strength of conventional concrete, the shear behavior and shear strength of reinforced recycled aggregate concrete beams with stirrups ratio of 0.14% and 0.19% were very similar to that of the corresponding reinforced conventional concrete beams [35]. By treating the recycled aggregate as two-phase material comprising residual mortar and natural aggregate, the shear performance of reinforced recycled aggregate concrete beams is comparable, or even superior, to that of beams made entirely with natural aggregate [36]. Generally, the tested recycled aggregate concrete beams could be designed conservatively by using the current codes for conventional concrete beam [31,33,34,35,36].

## 2. Research Significance

Based on the literatures reported [37], more researches on shear performance of reinforced recycled aggregate concrete beams without web reinforcement have been carried out and evaluated with the current design codes. Comparatively, less studies were performed on the shear behavior of reinforced recycled aggregate concrete beams with stirrups, thus further researches and experimental results are necessary. Consequently, to provide a quantitative basis for the safe application of CRAC, the shear performance, with equal importance to the bending performance, of reinforced CRAC beams should be determined.

This paper wants to bridge the gap of experimental study on the shear performance of reinforced CRAC beams with stirrups. Fourteen beams with stirrups were experimentally investigated under a four-point loading test. The main variables are the CRAC strength and the shear-span to depth ratio. The shear behaviors including concrete strain in shear-compression zone, stirrup strain, shear-cracking force, shear-crack width intersected with stirrups, crack distribution, and failure patterns are discussed based on the test results and the shear mechanisms. The equations used for predicting the cracking resistance and shear capacity of reinforced CRAC beams are proposed. Finally, the validity of standard design methods is evaluated.

## 3. Experimental Works

### 3.1. Raw Materials

As mentioned above, a three-stage-recycling-technology is used to produce recycled aggregate with rational morphology. In condition of the mother aggregate of demolished concrete with a maximum particle size of 20 mm, the recycled coarse aggregate was produced with a maximum particle size of 16 mm [11,27,28]. To get the maximum compact-closed density of coarse aggregates, 40% recycled coarse aggregate with particle size of 5–16 mm and 60% natural aggregate (crushed limestone) with particle size of 16–20 mm in weight of the total coarse aggregate were composited. This was also met the continuous grading of particles specified in China codes GB/T52177 and JGJ52 [15,24]. The fine aggregate was the byproduct of recycled coarse aggregate with particle size less than 5 mm after screening. The grading met the specification of sand for concrete in China code GB/T52176 [16]. The physical and mechanical properties of aggregates are presented in Table 1.

The ordinary silicate cement in strength grade of 42.5 and the class-II fly ash were used as binder. The physical and mechanical properties of cement are presented in Table 2. The fly ash was of 26% fineness modulus, passing through a 0.045 mm sieve, with a density of 2030 kg/m^3^, and a water demand ratio of 104%. Properties of the cement and fly ash were measured by using the standard methods in the lab. For CRAC of C50 and C60 strength grade, a portion of the cement was replaced by fly ash to adjust the workability of fresh CRAC. The additive was the polycarboxylic acid superplasticizer of PCA-I with 20% water reduction and a density of 520 kg/m^3^. The mix water was tap water.

### 3.2. Preparation of CRAC

The mix proportions of CRAC were designed with the absolute volume method and adjusted based on previous studies [11,27,28,38]. The dosages of aggregates were weighed in the condition of saturated dry surface. The additional water, computed as per the absorption of recycled aggregates, was added to presoak the recycled aggregates before mixing. Different dosages of additive were used in the mixture to keep the slump around 150 mm. After the trial mixing and adjustments, the mix proportions of CRAC are presented in Table 3.

The horizontal-shaft forced mixer made by Zhengzhou Jianyi co. LTD, Henan, China, was used to produce fresh CRAC. After pre-soaking the recycled aggregates for 1 h, the natural aggregate was added and mixed uniformly. Then the cement, fly ash and additive were mixed by adding the mixing water. The slump of fresh CRAC was measured (within 150 ± 20 mm) before casting to ensure the cast quality of all specimens.

### 3.3. Preparation of Reinforced CRAC Beams

Fourteen reinforced CRAC beams with normal rectangular section were designed, the sectional width *b* = 150 mm and depth *h* = 300 mm. The length was 2.7 m, the span *l*_0_ = 2.4 m. The shear-span was the center line distance from support to load application point. One trial of CRAC beams with the same strength grade of C40 was with the shear-span to depth ratio *λ* = 1.0, 1.5, 2.0 and 3.0. Another trial of CRAC with the same *λ* = 2.0 was with strength grade of C30, C40, C50 and C60. As per the specifications in China code GB50010 [25,26], two HRB500 hot-rolled deformed rebars were used as the longitudinal tensile reinforcement, with the diameter *d* = 20 mm, the measured yield strength *f*_y_ = 545 MPa, and the measured modulus of elasticity *E*_s_ = 2.05 × 10^5^ MPa. The thickness of concrete cover for longitudinal tensile reinforcement *c*_s_ = 25 mm. The longitudinal compression reinforcement was two HPB300 hot-rolled plain rebars with diameter of 12 mm. The stirrups were the HRB400 hot-rolled deformed rebars with the diameter *d*_sv_ = 6 mm and the sectional area of all legs of stirrups within spacing *A*_sv_=57 mm^2^, the measured yield strength *f*_yv_ = 460 MPa, and the measured modulus of elasticity *E*_sv_ = 2.10 × 10^5^ MPa. The spacing of stirrups was *s* = 150 mm and the designed stirrups ratio *ρ*_sv_ = *A*_sv_/*bs* = 0.253%. This met the design requirements of web reinforcement for conventional reinforced concrete beams *d*_sv_ ≥ 6 mm, *s* ≤ 150 mm and the minimum stirrups ratio *ρ*_sv,min_ = 0.24*f*_t_/*f*_yv_ = 0.129% (*f*_t_ is the tensile strength of C30 CRAC). Details of the beams are presented in Figure 1.

Two beams per group marked as a/b were manufactured for experimental study as per China code GB/T50152 [39]. The reinforcements skeleton was fixed into the steel mode to maintain the concrete cover and the placement during the casting of fresh CRAC. The CRAC was compacted by the slight vibration of the vibrators attached on the outside of steel mold. The vibrators are made by Xinxiang Hongda Vibration Equipment co. LTD, Henan, China. Accompanied with the beams in the same batch and curing condition, three cubes with dimension of 150 mm were produced to measure the splitting tensile strength (*f*_t_). Six prisms of 150 × 300 mm, three per group, were produced to measure the axial compressive strength (*f*_c_) and modulus of elasticity (*E*_c_) of CRAC as per China code GB/T50081 [40]. After casting and until demolding, the screed top-surface was covered by plastic film for 48 h. After demolding, the test beams and accompanied specimens were cured with sprayed water for 7 d, then placed in natural condition. After curing for 28 d, the loading tests were carried out. All the tests were finished within 10 d. 

### 3.4. Test Method

A four-point loading test was conducted by the load control as per China GB/T50152 [39]. Two concentrated loads were applied on top-surface of test beams by hydraulic jacks fixed on loading frames. The hydraulic jacks are made by Zhengzhou Jianyi co. LTD, Henan, China. The loads were measured by loading sensors and recorded automatically by an automatic data collection system. The loading sensors are made by Shanghai Donghua co. LTD. The shear-cracking force was determined as per initial shear crack with width less than 0.05 mm. The initial shear crack was a diagonal web-crack or a shear-flexural crack at bottom of shear-span turned to tilt to the load point. As presented in Figure 2, the strains of concrete in the shear-compression zone were measured by three strain gauges bonded on top and side surfaces of the beams. The strains of stirrups at points intersected with the connection between support and load point were measured by the strain gages bonded on the surface of stirrups. The shear crack width at the intersect points of stirrups was detected by a reading microscope with precision of 0.02 mm. The failure characteristics were set as the crushing of compressive CRAC or/and the abrupt widening of critical diagonal crack over 1.5 mm.

## 4. Test Results

### 4.1. Cracks and Failure Patterns

Figure 3 presents the crack distributions, critical cracks and failure patterns of test beams with different *λ*. 

For RLC40-1a and RLC40-1b with *λ* = 1, the first shear crack appeared on the web of shear-span. With the increase of loads, other shear cracks also occurred on the web of shear-span, and the parallel shear cracks formed with a greater crack width up to 0.9 mm. Finally, the compression strut of concrete on shear-span crushed and the critical diagonal crack was over 1.50 mm.

For RLC40-1.5a and RLC40-1.5b with *λ* = 1.5, the crack appeared initially at bottom surface and inclined afterwards to the loading point. The critical shear cracks occurred quickly almost along the connection between the support and loading point when the load was up to 60% of the ultimate. Finally, the beams failed when the concrete crushed in shear-compression zone, and the critical diagonal crack widening over 1.50 mm sharply.

For RLC40-2a and RLC40-2b with *λ* = 2, the shear crack appeared firstly at the shear-span near the loading point section, then with the increasing loads the second shear crack occurred near the support and inclined to the loading point. Two critical shear cracks formed in the shear-span and grew gradually. The beams failed with the critical diagonal crack widened over 1.50 mm sharply.

For RLC40-3a and RLC40-3b with *λ* = 3, the crack appeared initially with a greater diagonal angle on shear-span due to a bending-shear action near the loading point section. With the increase of loads, the second and third diagonal cracks occurred near the support and rapidly grew to be the critical cracks. The beams failed due to the critical cracks widened over 1.50 mm rapidly.

Based on above statements, the distribution of diagonal cracks and failure pattern of tested reinforced CRAC beams are controlled by the *λ*. These are similar to those of conventional reinforced concrete beams with the same bearing mechanisms as a result of the combination of bending and shearing actions. The beam fails with a feature of CRAC crush within an inclined column, CRAC crush at the shear-compression zone, and shear and diagonal tension when *λ* = 1, 1.5, 2 and 3, respectively.

Figure 4 exhibits the crack distributions, critical cracks and failure patterns of test beams with different CRAC strength and *λ* = 2. All beams had similar distribution of diagonal cracks on shear-span, while the beams with lower CRAC strength were more prone to bending crack near the loading point section at shear-span. This corresponds to the mechanical properties of CRAC with lower tensile strength than shear strength. The critical shear crack formed along the connection between loading point and support, and widened continously to over 1.5 mm as failure pattern. A little crushing of CRAC at shear-compression zone occurred on the beams RLC30-2b and RLC50-2a.

Table 4 presents the main test data of the sectional width (*b*) and depth (*h*), the effective depth *h*_0_ (=*h* − *c*_s_ − *d*/2), the axial compressive strength (*f*_c_), tensile strength (*f*_t_) and modulus of elasticity (*E*_c_) of CRAC, and the shear cracking force (*V*_cr_) and shear capacity (*V*_u_). 

### 4.2. Strains of CRAC and Stirrups

Figure 5 displays the strain variations of CRAC in shear-compression zone of test beams with different λ, in which the negative value represents the compressive strain. After the critical diagonal crack formed, the compressive strains of top surface within shear-span near loading points became into tension in cases with *λ* = 1 and 1.5, and tended to be reduced in cases with λ = 2 and 3. This is due to the vertical sectional shear deformation under the shear force acted on the top surface at the loading points. In cases with *λ* = 1, the top surface cracked under large tensile stress to create the compression strut. With the increase of the *λ*, the tensile effect decreased to maintain the top surface in compression. The compressive strains of CRAC measured by 1# and 2# gauges were compressive, and basically reduced along depth downward from the top surface at the same loads, and decreased with the increase of λ. This reflects that the entire compression zone depth at shear-span reduced with the increase of the λ, and the diagonal angle of critical crack became smaller and smaller. As a result, the load bearing pattern of CRAC at shear-span of beams trended to transfer from inclined compression, shear compression, shear to diagonal tension, and finally very small depth of uncracked block exists [41,42,43]. Therefore, corresponding to the failure patterns, this part of shear strength decreases with the increasing *λ*. 

Figure 6 presents the strain variations of CRAC in the shear-compression zone of test beams with different CRAC strength in cases with λ = 2. Similar changes of CRAC strain presented in shear-compression zone of the beams, while the larger compressive strain appeared on the beams with lower CRAC strength. This was due to the larger bending effect on deflection of the beam with smaller flexural stiffness accompanied with the decrease of CRAC strength [26,28]. However, the higher strength of CRAC in the shear-compression zone provides a higher resistance of the uncracked block under shear-compression at shear-span. This improves the shear resistance of CRAC which takes an important role in the bearing capacity of reinforced CRAC beams.

Figure 7 presents the tensile strain of stirrups at the intersection with critical diagonal crack. Based on the measured stirrup strain, and referring to the failure pattern presented in Figure 3; Figure 4, one stirrup was entirely activated by the critical diagonal crack in beams with λ = 1, two stirrups were entirely activated by the critical diagonal crack in beams with λ = 1.5 and 2, and three or four stirrups were entirely activated by the critical diagonal crack in beams with λ = 3. Accompanied with the sectional deformation, the stirrups acted as tensile bars to transfer the vertical forces across the diagonal cracks. The growth of stirrup strain was affected obviously by the λ. Except the slowest growth in the case with λ = 1 and the fastest growth in the case with λ = 3, the growths of stirrup strains are similar in cases with λ = 1.5 and 2 whatever the strength grade of CRAC changed from C30 to C60. This is related to the loading mechanisms of stirrups after the critical diagonal cracks formed on the beams. With the λ increased from 1 to 3, the main loading resistance of the beams transfer from CRAC to stirrups. Therefore, the continuous development of stirrup strain relies on the combined loading effect of the CRAC in shear-compression zone, and the stirrups intersected the critical diagonal cracks. While the development of tensile strain of stirrups decreased with the increase of CRAC strength under the same shear force. This is the same as the loading mechanisms of the conventional reinforced concrete beams in shear [26,41,42,43]. Referring to the real yield strain of stirrups *ε*_yv_ = *f*_yv_/*E*_sv_ = 2190με marked as the red vertical line in Figure 7, except the stirrups of the beams with λ=1 almost reached the yield, the others were all over the yield at failure state. This indicates that even if in the case with λ = 1, the stirrups are necessary to be used to transfer shear force within the governing direct strut of CRAC. Meanwhile, in the case with λ = 3, the tensile action of stirrups across the critical diagonal cracks controls the shear capacity. This means the only way is to increase the stirrup ratio to improve the bearing capacity in this condition. 

### 4.3. Shear-Cracking Resistance

As per the test results presented in Table 4, the shear-cracking force reduced with the increase of *λ*. This is similar to that of conventional reinforced concrete beams [26,44]. In cases with the same *λ*, the shear-cracking force increased with the strength of CRAC due to the increased tensile strength. The stirrups had slight benefit to the shear-cracking force which can be seen as the extra safety. Therefore, the shear-cracking force can be computed with the same equation as the conventional reinforced concrete beams, Equation (1) proposed by Zhao et al [44],
(1)$$V_{\mathrm{cr}}=\left(\frac{2.45}{\lambda +3.5}+\frac{20\rho }{\lambda +1.1}\right)f_{t}bh_{0}$$
where *ρ* is the longitudinal reinforcement ratio, *h*_0_ is the effective depth of cross section. In the second item, *λ* = 3 when *λ* > 3.

The Equation (2) is proposed by Rebeiz [45],
(2)$$V_{\mathrm{cr}}=\left[0.4+\sqrt{{f^{\prime }}_{c}\rho /\lambda }(2.7-0.4\alpha_{d})\right]bh_{0}$$
where *f*_c_’ is the cylindrical compressive strength of concrete; based on tests of conventional concrete in this paper *f*_c_’ = 0.81*f*_c_. *α*_d_ is the adjustment factor for shear-span to depth ratio; *α*_d_ = *λ* when 1.0 < *λ* < 2.5, *α*_d_ = 2.5 when *λ* ≥ 2.5.

Equation (3) is proposed by CEB-FIP MC 1990 [46],
(3)Vcr=0.15(3λ)13k(100ρf′c)13bh0
where *k* is the coefficient of sectional depth, $k=1+\sqrt{200/h_{0}}$. 

Figure 8 displays the comparison of tested to predicted values of shear-cracking force. The ratios of tested *V*_cr_ to predicted *V*_cr_ with Equations (1), (2) and (3) are in the range of 0.835~1.320, 0.841~1.124, and 0.963~1.464. The mean ratios are 0.994, 0.970, and 1.112, respectively, with a variation coefficient of 0.112, 0.083, and 0.121. Due to the inevitable difference of tested shear-cracking force determined in the condition of diagonal crack width ranged from 0.02 to 0.04 mm, the variation of the ratios is acceptable, and good prediction of shear-cracking force for reinforced CRAC beams can be given out.

### 4.4. Shear-Crack Width

The changes of maximum width of critical diagonal cracks intersected with stirrups are exhibited in Figure 9. This indicates a similar change of the maximum width of critical diagonal cracks compared with that of tensile strain of stirrups presented in Figure 7. This means that the crack width depends mainly on the elongation of stirrups under shear force. All the critical diagonal cracks reached the limit width of 1.5 mm as failure characteristic specified in China GB50010 [25]. 

### 4.5. Shear Capacity

Different theoretical models have been proposed for reinforced concrete beams without web reinforcements based on the analysis of the shear strength of the uncracked concrete in a compression zone, the interlocking action of aggregates along the cracked concrete surfaces on the sides of the crack, the dowel action of the longitudinal reinforcement, and the simultaneous occurrence of both arch action and beam action [26,41,42,43,47]. The shear contribution of stirrups depends on the action which bridges the intersected critical diagonal cracks [26,31,48]. Therefore, the shear capacity of the reinforced concrete beam with stirrups can be simply superposed by the shear strengths provided separately by the concrete and stirrups. Based on this principle, the equation proposed by statistical analysis of a large number of experimental results of reinforced concrete beams is used to predict the shear capacity of reinforced CRAC beams, as follows [48]:(4)$$V_{u}=\xi_{\lambda ,\rho }f_{t}bh_{0}+f_{\mathrm{yv}}A_{\mathrm{sv}}\frac{l_{h}}{s}h_{0}$$
(5)$$l_{h}=0.18+0.35\lambda$$
(6)$$\xi_{\lambda ,\rho }=\frac{0.115+0.192\lambda +28.7\rho }{\lambda -0.6}$$
where *ξ*_λ,*ρ*_ is the synthetical coefficient of all factors contributing to the shear capacity of CRAC; *ρ* is the longitudinal tensile reinforcement ratio; *l*_h_ is the horizontal projection length of critical diagonal cracks; *A*_sv_ is the sectional area of all legs of stirrups within spacing *s*. *λ* = 4.5 when *λ* ≥ 4.5, and *ρ* = 4.0% when *ρ* ≥ 4.0%.

From Equation (5), the calculated *l*_h_ is 0.53, 0.71, 0.88 and 1.23 for the beams with *λ* = 1, 1.5, 2 and 3. Correspondingly, the number of stirrups (*l*_h_*h*_0_/*s*) activated by critical shear crack is 0.95, 1.28, 1.56 and 2.21. This gives a conservative results of shear strength provided by stirrups. However, good predictions are given out with the statistical results of the shear capacity of test beams, as presented in Figure 10, except a higher value for the beams with *λ* = 3. This is due to the overestimation of shear strength contributed by stirrups with yield strength of 460 MPa. Although the stirrups reached the yield strength at failure of the beams with *λ* = 3, less shear force increased with the fast growth of stirrup strain as presented in Figure 7. With the increase of the *λ* from 1 to 3, as presented in Figure 10b), the percent of shear strength provided by CRAC reduces from 89% to 46%, in opposite the shear strength of stirrups raises from 11% to 54%. This is about 25% overestimated shear strength compared with the 335 MPa stirrups in conventional reinforced concrete beams. If the stirrups strength is taken as the critical value of 400 MPa as per China code GB50010 [25] or 413 MPa as per ACI318-19 [49], the predictions become better.

## 5. Synthetical Discussion

For the design of the shear performance of reinforced concrete beams, the control of shear crack at normal service state is contained in the design of bearing capacity. The first measure is to reduce the value of yield strength of stirrups to be factored strength considering a safety coefficient on the basis of critical yield strength. Due to the yield of stirrups intersecting the critical diagonal cracks, the shear force subjected by certain stirrups is constant. The shear resistance provided by stirrups can be predicted as per China code GB50010 and ACI318-19 [25,49],
(7)$$V_{\mathrm{sv}}=f_{\mathrm{yv}}\frac{A_{\mathrm{sv}}}{s}h_{0}$$

In this study, the critical yield strength of HRB400 rebar is 400 MPa. Taking the safety coefficient as 1.1, the factored strength of stirrups is 360 MPa. Corresponding to the factored strength, the tensile strain of stirrups *ε*_sv_ = *f*_y_/*E*_sv_ =360 MPa/1715 με is marked as the black vertical line in Figure 7. It can be seen that a sufficient safety has put on the stirrups to subject the shear force.

Then, the shear resistance attributed to the CRAC can be calculated from Equation (8). The predictive equations as per China code GB50010 and ACI318-19 are expressed as Equations (9) and (10), respectively.
(8)$$V_{u}=V_{c}+V_{\mathrm{sv}}$$
(9)$$V_{c}=\frac{1.75}{\lambda +1}f_{t}bh_{0}$$
where *λ* = 1.5 when *λ* < 1.5; *λ* = 3 when *λ* > 3.
(10)$$V_{c}=0.17\sqrt{{f^{\prime }}_{c}}bh_{0}$$

As per ACI318-19, the total shear capacity of reinforced concrete beams should satisfy the limit as presented in Equation (11),
(11)$$V_{u}\leq 0.83\sqrt{{f^{\prime }}_{c}}bh_{0}$$

In the obsolete Germany code DIN-1045-1-2008 [50], the shear capacity of reinforced concrete beams is also expressed as Equation (8). The shear resistance provided by stirrups is considered 1.1 times that of Equation (7), as expressed in Equation (12), and the shear resistance provided by CRAC is calculated with Equation (13):(12)$$V_{\mathrm{sv}}=1.1\frac{A_{sv}f_{yv}}{s_{}}h_{0}$$
(13)$$V_{c}=0.43{f^{\prime }}_{c}^{1/3}bh_{0}$$

Meanwhile, in Model Code 2010 [51], at Level III approximation, the shear capacity of reinforced concrete beams with stirrups can be calculated with the following Equations (14)–(22),
(14)$$V_{u}=V_{c}+V_{sv}\leq V_{u,\max}$$
(15)$$V_{c}=k_{v}\sqrt{{f^{\prime }}_{c}}bz$$
(16)$$V_{\mathrm{sv}}=\frac{A_{sv}f_{yv}}{s_{}}z\cot\theta$$
(17)$$V_{u,\max}=\eta_{\mathrm{fc}}{f^{\prime }}_{c}bz\cos\theta_{\min}\sin\theta_{\min}$$
(18)$$k_{v}=\frac{0.4}{1+1500\varepsilon_{x}}\left(1-\frac{V}{V_{u,\min}}\right)$$
(19)$$\varepsilon_{x}=\frac{1}{2E_{s}A_{s}}\left(V+\frac{Va}{z}\right)$$
(20)ηfc=[30fc]1/3≤1.0
(21)$$\theta_{\min}=20{}^{\circ}+10000\varepsilon_{x}$$
(22)$$\theta_{\min}\leq \theta \leq 45{}^{\circ}$$

Taking *V* = *V*_u_ for Equations (18) and (19), *z* = 0.9*h*_0_. The shear resistances attributed to the CRAC and provided by stirrups can be determined.

Substituting the tested data of this study into above equations, the tested shear resistance attributed to the CRAC can be obtained for each beams by the tested shear capacity minus the calculated shear resistance provided by stirrups. Ratios of $V_{c}/f_{t}bh_{0}$, $V_{c}/\sqrt{{f^{\prime }}_{c}}bh_{0}$ and $V_{c}/{f^{\prime }}_{c}^{1/3}bh_{0}$ are computed and exhibited as scatter plots in Figure 11. Compared with the lower values of these ratios calculated by Equations (9), (10), (13) and (15), the tested shear capacities of reinforced CRAC beams are higher than the predicted by design Equations specified in GB50010, ACI318-19 and Model Code 2010, except that DIN-1045-1-2008 gives a higher resistance of beams with *λ* = 3. China code GB50010 adopts a lower limit changed with the *λ*, while ACI318-19 and DIN-1045-1-2008 adopt a constant lower limit. Comparatively, Model Code 2010 gives a much lower shear resistance attributed to the CRAC.

The utilization level of the shear resistance attributed to the CRAC in above design codes can be got by dividing calculated values with tested ones, as presented in Figure 12. Model Code 2010 gives a largest safety with the utilization level increased from 15.6% to 60.6% in case of the *λ* varying from 1 to 3. ACI 318-19 gives a similar safety with the utilization level increased from 16.3% to 66.3%. DIN-1045-1-2008 has a higher safety with the utilization level increased from 25.0% to 80.0% in case of the *λ* varying from 1 to 2.5, compared with the utilization level of GB50010 increased from 38.5% to 80.0%. However, DIN-1045-1-2008 exists a lower safety in case of the *λ* over than 2.5 with higher utilization level until the safety is consumed at *λ* = 3. Comparatively, China code GB50010 rationally uses the shear strength attributed to the CRAC at a relatively higher level. This benefits to developing the beneficial contribution of CRAC.

Generally, the ratio of tested to predicted shear capacity of reinforced CRAC beams in this study by using the Equations of China code GB50010, ACI318-19, Model Code 2010 and DIN-1045-1-2008 ranges within 1.778–1.040, 3.405–1.241, 3.471–1.215 and 2.637–0.960 respectively. The prediction of shear capacity of reinforced CRAC beams are conservative by using the design equations of the former three design codes, except that DIN-1045-1-2008 gives a higher prediction for beams with *λ* = 3. This is consistent to above discussion about the utilization level of CRAC. 

However, due to the higher utilization of CRAC subjected to shear action, the shear safety of reinforced CRAC beams with *λ* = 3 relies much more on the reinforcement of stirrups. As per China code GB50010, the calculated shear force corresponding to bending failure is 96.8 kN. This means that the diagonal tension was accompanied by a composite action of large longitudinal tensile stress expressed by Equation (19) [51]. Therefore, the stirrups fast elongated after the formation of critical diagonal cracks, and the crack width was fast widening with less increment of shear force. 

In engineering structures, the load factor is considered to differentiate the serviceability limit state and the ultimate limit state [25,26]. Taking the load factor as 1.35, the unfactored shear force on the beams at serviceability limit state can be approximately back computed with the shear capacity calculated by using the equations of design codes. That is, the unfactored shear force of *V*_u_/1.35 is supposed to act on the beams at serviceability limit state. Combined with Figure 9, the maximum width of diagonal cracks (*w*_s,max_) on the beams at normal service can be extracted. As presented in Table 5, the maximum width of diagonal cracks can be controlled within 0.15 mm for reinforced CRAC beams when ACI 318-19 and Model Code 2010 are followed. A larger diagonal crack width exists on the reinforced CRAC beams when China code GB 50010 is followed. This is a result without considering the safety factor of CRAC.

## 6. Conclusions

Based on the experimental study of shear behaviors of reinforced CRAC beams, the conclusions can be drawn as follow:

(1) The diagonal cracking, crack distribution and failure pattern of reinforced CRAC beams depended on the shear-span to depth ratio, and were slightly affected by the CRAC strength. The changes of CRAC strains in shear-compression zone and stirrup strains accompanied with the appearance and extension of diagonal cracks reflected the tendencies of shear strength provided by CRAC and stirrups in different loading periods. The stirrups yielded at intersected section to critical diagonal cracks when the failure of test beams took place. With the *λ* increased from 1 to 3, the failure patterns of test beams appeared as CRAC crushing within a slant column, CRAC crushing at shear-compression zone, shear with continuous widening of diagonal cracks, and diagonal tension with fast widening of critical diagonal crack.

(2) The shear-cracking resistance and shear capacity of reinforced CRAC beams reduced with the increase of shear-span to depth ratio, and increased with the CRAC strength. It can be predicted well by the statistical equations used for conventional reinforced concrete beams.

(3) The shear strengths provided by CRAC and stirrups are analyzed based on the design equations specified in China code GB50010, ACI318-19, Model Code 2010 and Germany DIN-1045-1-2008. The factored strength of HRB400 rebar is rational to be 360 MPa for the calculation of shear strength provided by stirrups. The utilization level of CRAC raises with a sequence when Model Code 2010, ACI318-19 and China code GB50010 are followed. As specified in China code GB50010, a rational utilization of CRAC is possible by using the shear resistance attributed to the CRAC changed with the shear-span to depth ratio. The predictions of shear capacity of reinforced CRAC beams are conservative as per China code GB50010, ACI318-19 and Model Code 2010.

## Figures and Tables

**Figure 1 materials-13-01711-f001:**
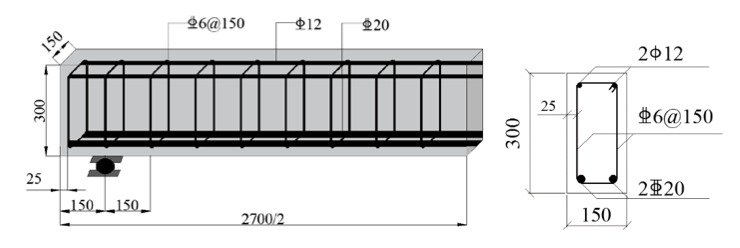
Details of test beams (unit: mm).

**Figure 2 materials-13-01711-f002:**
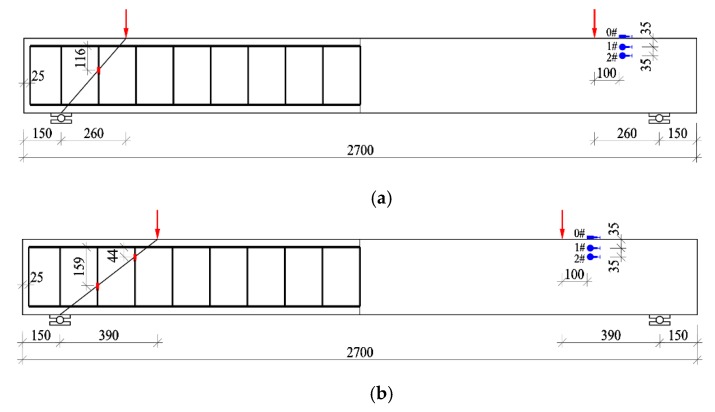
Arrangement for strain gauges of concrete and stirrups: (**a**) *λ* = 1; (**b**) *λ* = 1.5; (**c**) *λ* = 2; (**d**) *λ* = 3.

**Figure 3 materials-13-01711-f003:**
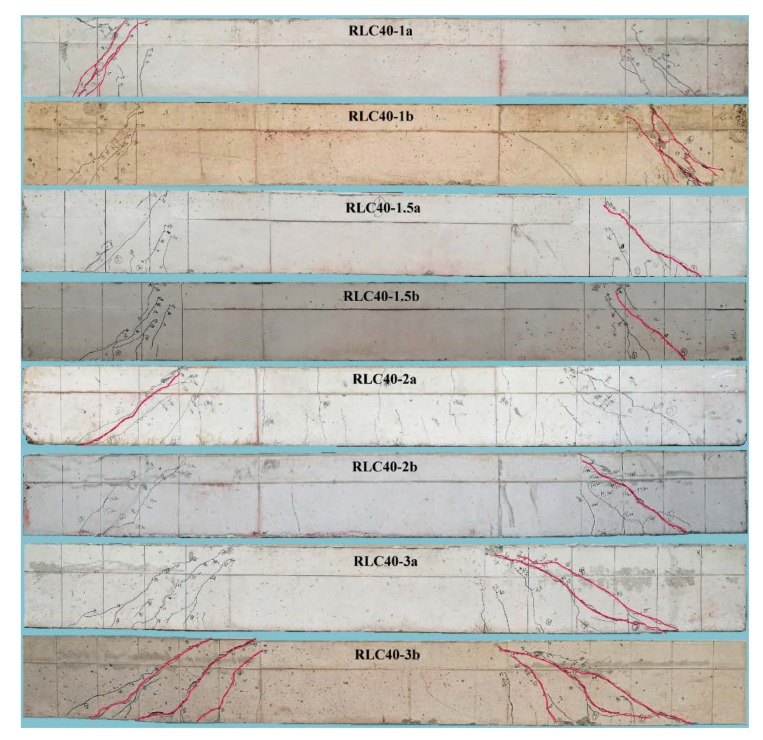
Crack distribution and failure pattern of test beams with different *λ*.

**Figure 4 materials-13-01711-f004:**
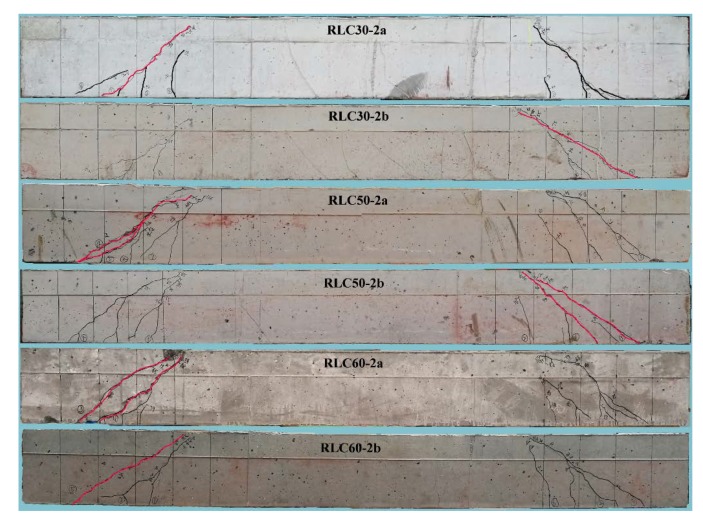
Crack distribution and failure pattern of test beams with different CRAC strength.

**Figure 5 materials-13-01711-f005:**
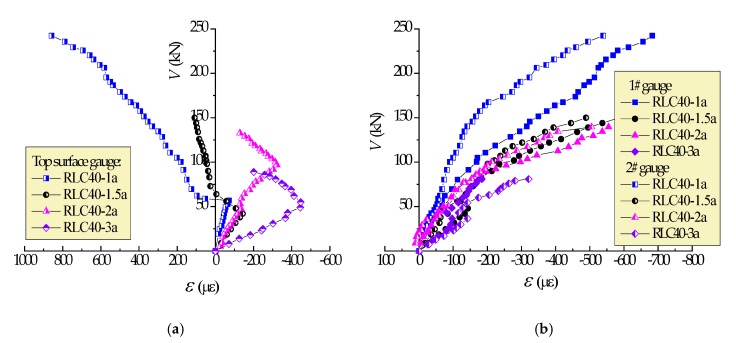
Concrete strain in shear-compression zone of test beams with different λ: (**a**) top surface gauge; (**b**) 1# and 2# gauges on side surface.

**Figure 6 materials-13-01711-f006:**
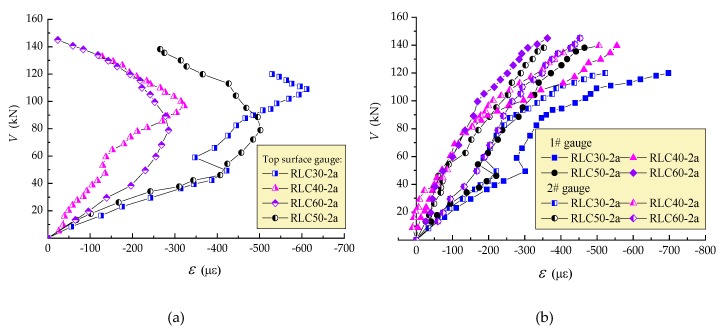
Concrete strain in compression-shear region of test beams with different strength of CRAC: (**a**) top surface gauge; (**b**) 1# and 2# gauges on side surface.

**Figure 7 materials-13-01711-f007:**
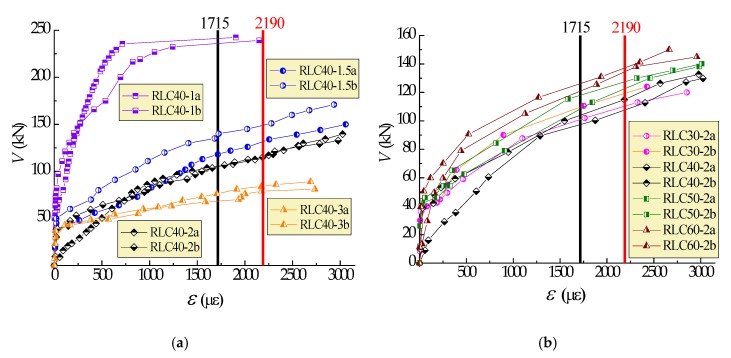
Strain of stirrups intersected critical diagnol cracks of test beams: (**a**) λ = 1~3; (**b**) C30~C60.

**Figure 8 materials-13-01711-f008:**
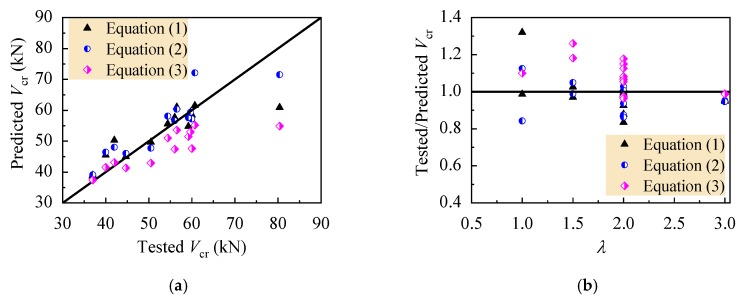
Comparison of tested with predicted shear-cracking force: (**a**) values; (**b**) ratios.

**Figure 9 materials-13-01711-f009:**
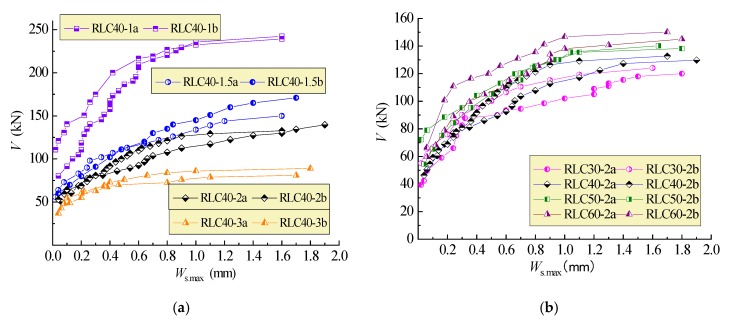
Cracks at points of stirrups intersecting shear-cracks of test beams: (**a**) λ = 1~3; (**b**) C30~C60.

**Figure 10 materials-13-01711-f010:**
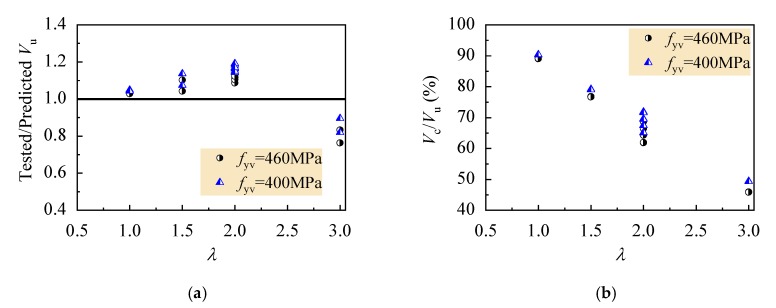
Statistical results of bearing capacity of test beams: (**a**) ratios of tested to predicted values; (**b**) percent of shear strength provided by CRAC.

**Figure 11 materials-13-01711-f011:**
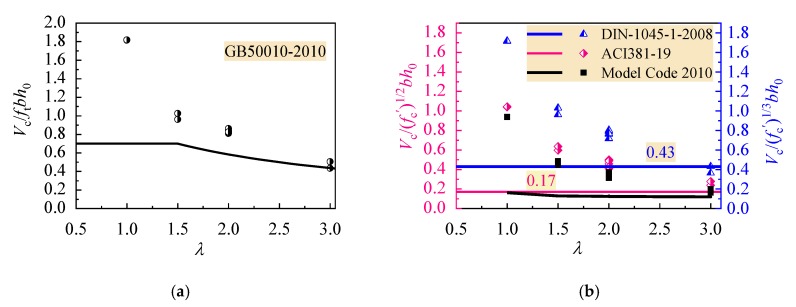
Comparison of shear strength attributed to the CRAC with current design codes: (**a**) China code GB50010; (**b**) ACI318-19 and DIN-1045-1-2008.

**Figure 12 materials-13-01711-f012:**
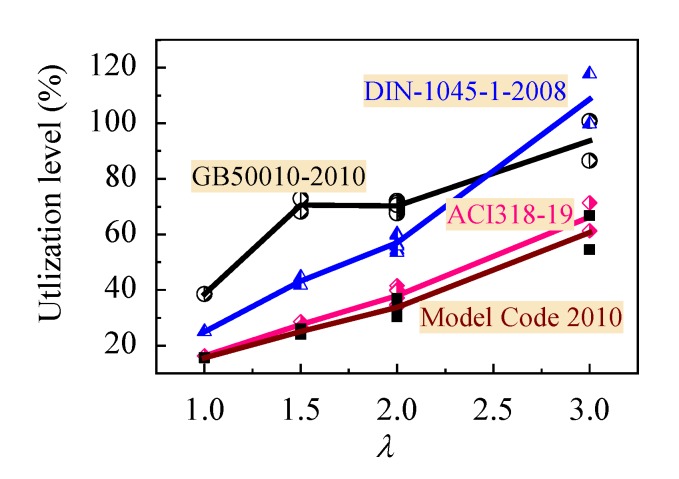
Utilization level of the shear strength attributed to the CRAC in design codes.

**Table 1 materials-13-01711-t001:** Physical and mechanical properties of fine and coarse aggregates.

Properties	Coarse Aggregate		Recycled Fine Aggregate
	Natural	Recycled	
Apparent density (kg/m^3^)	2722	2674	2396
Bulk density (kg/m^3^)	1417	1293	1330
Compact-closed density (kg/m^3^)	1592	1445	1470
Water absorption at 24 h (%)	0.47	5.1	9.5
Crush index (%)	12.8	14.3	-
Fineness modulus	-	-	3.28

**Table 2 materials-13-01711-t002:** Physical and mechanical properties of cement.

Grade	Density (kg/m^3^)	Consistency	Setting Time (min)		Compressive Strength (MPa)		Flexural Strength (MPa)	
			Initial	Final	3d	28d	3d	28d
42.5	3060	27.6	160	265	28.6	45.8	4.5	5.7

**Table 3 materials-13-01711-t003:** Mix proportion of composite-recycled aggregate concrete (CRAC).

Mix identifier	C30	C40	C50	C60
Water/binder ratio	0.55	0.44	0.28	0.24
Water (kg/m^3^)	200	200	175	165
Cement (kg/m^3^)	362.8	455.7	562.5	584.4
Fly ash (kg/m^3^)	-	-	62.5	103.1
Recycled fine aggregate: 0–5 mm (kg/m^3^)	725.4	693.6	663.8	648.8
Recycled coarse aggregate: 5–16 mm (kg/m^3^)	550.9	526.7	504.1	412.5
Natural coarse aggregate: 16–20 mm (kg/m^3^)	450.8	431.0	412.5	403.2
Additional water (kg/m^3^)	51.0	48.8	39.8	38.9
Additive (kg/m^3^)	0	1.82	6.25	8.94

**Table 4 materials-13-01711-t004:** Main test data of reinforced CRAC beams.

Beam ID.	*b* (mm)	*h* (mm)	*h*_0_ (mm)	*λ*	*f*_c_ (MPa)	*f*_t_ (MPa)	*E*_c_ (GPa)	*V*_cr_ (kN)	*V*_u_ (kN)
RLC40-1a	150	312	277	1	27.5	2.71	31.2	60.7	242.3
RLC40-1b	150	308	273	1	27.5	2.71	31.2	80.4	239.2
RLC40-1.5a	150	306	271	1.5	26.8	2.89	32.3	56.0	150.0
RLC40-1.5b	150	310	275	1.5	26.8	2.89	32.3	60.0	171.0
RLC40-2a	150	305	270	2	26.6	2.75	31.1	50.5	139.6
RLC40-2b	150	308	273	2	26.6	2.75	31.1	42.0	132.7
RLC40-3a	150	302	267	3	26.8	2.56	32.7	37.0	81.0
RLC40-3b	150	304	269	3	26.8	2.56	32.7	37.0	89.0
RLC30-2a	150	309	274	2	23.7	2.48	30.6	40.0	120.0
RLC30-2b	150	306	271	2	23.7	2.48	30.6	44.7	124.1
RLC50-2a	150	305	270	2	46.1	3.03	35.7	59.2	138.2
RLC50-2b	150	309	274	2	46.1	3.03	35.7	54.4	140.2
RLC60-2a	150	300	265	2	51.6	3.35	36.8	59.7	145.0
RLC60-2b	150	307	272	2	51.6	3.35	36.8	56.5	150.1

**Table 5 materials-13-01711-t005:** Maximum width of diagonal cracks on test beams at normal service state.

Beam ID	GB50010		ACI318-19		Model Code 2010	
	*V*_u_/1.35 (kN)	*w*_s,max_ (mm)	*V*_u_/1.35 (kN)	*w*_s,max_ (mm)	*V*_u_/1.35 (kN)	*w*_s,max_ (mm)
RLC40-1a	101.0	0.14	52.7	-	51.7	-
RLC40-1b	99.6	0.09	52.0	-	51.1	-
RLC40-1.5a	88.4	0.23	51.3	-	59.4	0.03
RLC40-1.5b	89.6	0.28	52.0	-	60.2	0.04
RLC40-2a	75.5	0.20	51.0	0.06	57.4	0.10
RLC40-2b	76.3	0.25	51.6	0.06	57.9	0.07
RLC40-3a	60.3	0.20	50.5	0.08	51.7	0.10
RLC40-3b	60.7	0.21	50.9	0.12	51.9	0.14
RLC30-2a	71.8	0.25	50.4	0.07	57.0	0.15
RLC30-2b	71.0	0.18	49.9	0.02	56.6	0.04
RLC50-2a	80.4	0.08	58.5	-	62.2	0.04
RLC50-2b	81.6	0.20	59.4	0.05	62.8	0.07
RLC60-2a	84.4	0.20	59.2	0.02	62.6	0.08
RLC60-2b	86.6	0.16	60.7	0.04	63.7	0.06

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
