# Peer review of "Shear Performance of Reinforced Concrete Beams Affected by Satisfactory Composite-Recycled Aggregates"

_materials, 2020, doi:10.3390/ma13071711_

Round 1

Reviewer 1 Report

This article is about an important problem. The research is well described and the conclusions are rational. My remarks:
1. Too detailed (and somewhat too long) abstract.
2. Introduction and Research significance. The style and grammar of some sentences should be improved. The authors cited 55 publications. This is far too much, the more so because many of them are not directly related to the presented research.
3. Many citations of the authors' works were used. Respecting the indisputable achievements of the authors in this subject, I believe that this is morally questionable, and this is not necessary for a full understanding of the article.

Author Response

Dear professor,

Thanks for your reviewing my paper. The response to your comments are stated as follow:

  1. Too detailed (and somewhat too long) abstract.
    Replay: Thanks. The abstract has been shortened.
  2. Introduction and Research significance. The style and grammar of some sentences should be improved. The authors cited 55 publications. This is far too much, the more so because many of them are not directly related to the presented research.
    Replay: Thanks. The introduction and research significance have been corrected. The references have been reselected to closely relate to the topic of this paper.
  3. Many citations of the authors' works were used. Respecting the indisputable achievements of the authors in this subject, I believe that this is morally questionable, and this is not necessary for a full understanding of the article.

Replay: Thanks. The authors list their papers trying to explain development process of composite-recycled aggregate concrete by themselves. Respecting your comment, we delete 6 papers seemed not closely to the presented study. Please check it.

However, respecting to other reviewer’ comment, five papers are added as references in revised manuscript.

Additionally, the grammar and style of sentences you mentioned in PDF document have been corrected. Please check them.

Thanks again,

Best regards,

Changyong Li

Reviewer 2 Report

The paper presents an experimental programme focusing on the shear response of composite recycled aggregate concrete, tested under four-point bending and with varying shear span-to-depth ratios. Additionally, the authors study the adequacy of existing codified procedures. The paper has some interesting technical content but needs significant modifications in order to be considered for publication. Although the results may be valuable and useful for the readers of the journal and the manuscript is relatively well organised, parts of the paper are non-readable due to language errors.

  • The abstract is too long and has very long phrases without connection. It needs to be completely rewritten in a more concise manner by clearly highlighting the novelty of the study and some of the key results. Never use 4-5 lines sentences, particularly when the quality of English is poor. Sentences from P1L19-22, P1L23-26, P128-32 are too long. Break them in shorter sentences of 1 or 2 lines.
  • Authors need to compare their results to up to date design codes. A newer version of ACI318 exists (i.e. issued in 2019) and the German DIN-1045-1-2008 is obsolete. The code comparisons need to be carried out against ACI318-19 and the European design code Eurocode 2, rather than DIN-1045. Additionally, authors can consider comparisons with Model Code 2010, which is the predecessor of the new Eurocodes (2020+).
  • Replace the following: P1L17 ‘changed’ with ‘was between’, P1L25 add ‘are’ controlled, and add ‘are’ slightly..
  • The literature review is very short and reports a limited amount of studies from other authors. Avoid lumping references at the end of the paragraphs and explain why each reference is relevant for the paper. This is extremely necessary for refs 10,14,15,23-30. Authors are suggested to expand the literature review as suggested above and keep only relevant references. Papers on steel fibre concrete do not seem to be relevant for this study. A cursory look into the literature shows that there are a series of experimental studies on RAC beams in shear (e.g. for your consideration: doi.org:10.1680/macr.2008.62.2.103; doi.org:10.1016/j.engstruct.2017.03.026; doi.org:10.1617/s11527-006-9161-5; doi.org:10.1016/j.conbuildmat.2016.08.034; doi.org:10.1016/j.conbuildmat.2013.12.019; doi.org:10.1016/j.engstruct.2017.05.028; doi.org:10.1016/j.conbuildmat.2016.10.058; doi.org:10.1016/j.conbuildmat.2016.08.022; doi.org:10.1016/j.conbuildmat.2018.10.023).
  • Rewrite the phrase from P2L48-51 by breaking in 3-4 short sentences, as it is unclear.
  • P2L60-72 the content of this paragraph reports the recycling methodology and has to be moved to the Raw Materials section. Replace this paragraph with an in-depth literature review.
  • P2L79 – please specify which are the ‘shortcomings of recycled coarse aggregate’…
  • P2L90 – please expand/clarify which are the ‘loading mechanisms of CRAC’?
  • P2L95-95; P3L101-103; P3L105-109 – please rewrite/break in several sentences – the current form of the sentence is unreadable
  • P3L112 – ‘tends to be the same.’ – please add the relevant references
  • P3L115 – ‘codes compared.’ – please add the relevant reference
  • P3L113 – changed in ‘a’ similar ‘manner’ – please correct
  • Research significance – the authors need to expand the literature review and clarify the novelty of the study; or clearly highlight the limitations of past studies and present what is new in the paper submitted.
  • P3L128 – replace ‘composited’ with ‘considered’.
  • P3L130 – describe more in detail the methodology of obtaining the CRAC – possibly move the paragraph from the introduction (P2L60-72) in this section
  • P3L126 onwards – please report the specific gravity, water absorption and possibly the fineness modulus of natural and recycled aggregates
  • P3L138 – please clarify whether the values in Table 2 are from manufacturer datasheet or assessed in the lab; please report if the data regarding the fly ash and admixtures are from datasheets or elsewhere
  • P4L149 – ‘saturated dry surface’ – saturated and dry seem to be contradicting terms; please clarify how saturation was achieved and verified?
  • P4L150 – correct to ‘pre-soaked’
  • Table 3 – add size ranges for the recycled and natural aggregates
  • P4L157 – clarify which was the slump of the investigated mixes or report the standard deviation form ‘150mm’ as reported in P4L151
  • P4L162 – replace ‘changed’ with ‘was’
  • P4L162 – how is the shear span-to-depth ratio measured? Were the sizes of the loading plates considered? Or was it centre line support to centre line of load application point?
  • P4L167 – replace ‘construction’ with ‘compression’
  • P4L169 – was the yield strength assessed by means of tensile coupon tests? How many samples were tested? Or is this the nominal yield strength
  • P5L177 – replace ‘mode’ with ‘mould’
  • P5L177-179 – unclear sentence – rephrase or break in shorter sentences
  • P5L184 – replace ‘demoulded’ with ‘demoulding’
  • P5L186 – replace ‘real’ with ‘measured’ (note: both nominal and measured sizes are real)
  • Table 4 – the values of elastic modulus Ec are not correct; they should be in the range of 30 GPa – please correct
  • Table 4 – please add the standard deviation of the compressive fc and tensile ft strengths
  • Table 4 – add a note to the table with all notations
  • P5L181 – replace ‘exerted’ with ‘applied’
  • P5L193 – please clarify how the experimental/test Vcr was assessed?
  • P5195-196 – these are test results – please move the sentence to the ‘results’ section
  • P5L201 – replace ‘precise’ with ‘precision’ or ‘accuracy’
  • P5/Section 2.4 – please add a figure with the position of the strain gauges on the rebars and concrete.
  • P5/Section 5 – Before start reporting the crack kinematics and failure patterns, the authors must report the load-deflection curves and show how the shear-span to depth ratios have influence the strength, stiffness and failure mode.
  • P5/Section 5 – Please do report in this section or/and Table 5 the flexural strengths (Vflex) of the beams. This is essential to understand the shift in failure modes from diagonal strut crushing to possible mixed-mode shear-bending.
  • Figures 3 and 4 – there can be only a single critical/failure crack – please identify the critical crack on the images (also correct P7L233)
  • P7L218 & P7L222 – replace ‘turned up’ with ‘developed’ or ‘occurred’
  • P7L219 – replace ‘inclined column’ with ‘compression strut’!
  • P7L233 – replace ‘sustainably’ with ‘gradually’
  • P8L241 – replace ‘was’ with ‘were’
  • P8L242 – ‘shear-span’ not ‘spen’
  • P8L245 – crushing
  • P8L254-255 – replace ‘was teared’ with ‘cracked’, ‘inclined column’ with ‘inclined strut’
  • P8L261 – clarify what ‘no depth of uncracked block exists’ (add a drawing to explain if possible)
  • P8L268 – consider using synonyms as ‘illustrated/depicts/presents’ to ‘displays’
  • P9L272 – ‘flexural stiffness …’ where is this shown for the tests reported in this paper? Please expand/clarify
  • P9L280-291 – please clarify how many stirrups were provided with strain gauges – please add a figure to clarify
  • P9L280-291 – how many stirrups have been activated by the diagonal crack? Please add a table or diagram (please see and analyse doi:10.1016/j.engstruct.2016.03.022 and doi:10.1680/jmacr.15.00405). This is essential to understand which is the contribution of Vs to the total shear capacity)
  • P10L294 – replace ‘crushed slant columns’ to ‘governing direct strut’
  • P10L305 – replace ‘security’ with ‘safety’
  • P10L314 – how the cylinder-to-cube strength conversion was determined? This generally varies with the concrete strength
  • P10L314 – alpha_d is an adjustment factor for what? Please clarify
  • P10L322 – replace ‘differential’ with ‘difference’
  • P11L345 – ‘unchanged mechanisms as use of CRAC’ – could the authors expand whether the shear transfer actions available in conventional concrete would have a similar contribution, particularly the tensile residual stresses and aggregate interlock. Just a thought - CRAC seem to have lower ft and different aggregate shapes compared with conventional cases. Please see and compare qualitatively with doi:10.1016/j.engstruct.2015.07.033, doi:10.1680/macr.12.00142)
  • P11/Section 3.5 – Shear capacity – Design codes consider a shear strength reduction factor (e.g Eurocode 2), or inversely a shear enhancement factor (e.g. British Standards), when the shear capacity is assessed for members loaded within relatively low shear span-to-depth ratios (e.g. a/d<2 or 2.5). Could the authors update their calculations by accounting such a factor – this was the utilisation level from Figure 12 would have a different representation (possibly more uniform). The way Figure 12 is presented may be misleading.
  • P14L422-425 – Please rephrase – sentence unclear
  • P14L426-433 – What is the purpose of this paragraph? Please clarify
  • P14/Section 5 – Conclusions – please make sure that all changes within the manuscript are reflected in this section and in the abstract. Also please aim at having a more concise Conclusions section with shorter sentences. As the case for the most length of the manuscript, this is very hard to read.
  • General note - please aim at proof-reading thoroughly the manuscript before re-submission and highlight all changes in colour within the revised manuscript to avoid having two rounds of revisions.

Round 2

Reviewer 2 Report

Thank you for addressing the comments and recommendations.